# CAD-RADS Scoring using Deep Learning and Task-Specific Centerline Labeling

**Felix Denzinger**[1,2]                                                    FELIX.DENZINGER@FAU.DE
[1] *Pattern Recognition Lab, Friedrich-Alexander-Universität Erlangen-Nürnberg, Erlangen, Germany*
[2] *Siemens Healthcare GmbH, Computed Tomography, Forchheim, Germany*

**Michael Wels**[2]
**Oliver Taubmann**[2]
**Mehmet A. Gülsün**[2]
**Max Schöbinger**[2]
**Florian André**[3]
[3] *Das Radiologische Zentrum - Radiology Center, Sinsheim-Eberbach-Erbach-Walldorf-Heidelberg, Germany*

**Sebastian J. Buss**[3]
**Johannes Görich**[3]
**Michael Sühling**[2]
**Andreas Maier**[1]
**Katharina Breininger**[4]
[4] *Department Artificial Intelligence in Biomedical Engineering, Friedrich-Alexander-Universität Erlangen-Nürnberg, Erlangen, Germany*

## Abstract

With coronary artery disease (CAD) persisting to be one of the leading causes of death worldwide, interest in supporting physicians with algorithms to speed up and improve diagnosis is high. In clinical practice, the severeness of CAD is often assessed with a coronary CT angiography (CCTA) scan and manually graded with the CAD-Reporting and Data System (CAD-RADS) score. The clinical questions this score assesses are whether patients have CAD or not (rule-out) and whether they have severe CAD or not (hold-out). In this work, we reach new state-of-the-art performance for automatic CAD-RADS scoring. We propose using severity-based label encoding, test time augmentation (TTA) and model ensembling for a task-specific deep learning architecture. Furthermore, we introduce a novel task- and model-specific, heuristic coronary segment labeling, which subdivides coronary trees into consistent parts across patients. It is fast, robust, and easy to implement. We were able to raise the previously reported area under the receiver operating characteristic curve (AUC) from 0.914 to **0.942** in the rule-out and from 0.921 to **0.950** in the hold-out task respectively.

**Keywords:** Coronary Artery Disease, Coronary CT Angiography, Deep Learning, Ensembling, CAD-RADS, Coronary Artery Labeling

## 1. Introduction

Worldwide, coronary artery disease (CAD) still is the leading cause of death (Roth et al., 2020), thus impacting the lives of many. Therefore, developing algorithms to support physicians with the diagnosis is of high interest. These algorithms may serve as a second reader

to ensure that no aspect is missed or to point the physician to areas of interest, thus speeding up the workflow.

CAD is predominantly linked to atherosclerotic plaque deposits aggregating within the vessel wall (Fuster et al., 1992). The degree of vessel narrowing – also called stenosis – caused by such a plaque deposit is an essential piece of information regarding patient risk and can be obtained using a coronary CT angiography (CCTA) scan. To report findings, assess patients' general condition, and to guide the clinical workflow the coronary artery disease-reporting and diagnosis system (CAD-RADS) score was introduced (Cury et al., 2016). This score is usually determined through a manual assessment by a human reader scoring the whole coronary vessel tree. It consists of six grades ranging from 0 to 5, where 0 refers to "no CAD present", 1-2 to "non-obstructive CAD present" and 3-5 to "obstructive CAD present", with a rising severeness within this grouping. Hence, primary clinical questions of interest are whether patients do have CAD or not (rule-out) and whether they suffer from obstructive CAD and therefore should undergo further (invasive) assessment including potential immediate revascularization or not (hold-out). However, this manual grading is time-consuming and reader/experience dependent (Razek et al., 2018; Maroules et al., 2018; Hu et al., 2021). Therefore, introducing decision support algorithms for this task is of high interest.As related work regarding this task is sparse, we discuss work on the related task of predicting severe stenosis degree. Algorithms performing this task can be divided into lesion-wise, and branch-wise.Lesion-wise algorithms focus mainly on the task of detecting and (separately) scoring one or multiple plaque deposits within the whole coronary vessel tree. Most of these approaches work on multi-planar reformatted (MPR) volumes created by interpolating orthogonal planes for each vessel centerline point. Commonly, these approaches are based on recurrent convolutional neural networks (RCNN) (Zreik et al., 2018; Denzinger et al., 2019; Ma et al., 2021). For these, a series of overlapping cubes along the centerline dimension is used, from which spatial features are extracted using a 3D convolutional neural network (CNN) at each position. The resulting feature sequence is analyzed using a recurrent neural network (RNN) (Zreik et al., 2018) or combined using a transformer module (Ma et al., 2021). A branch-wise approach presented by (Candemir et al., 2020) utilizes a 3D CNN which takes whole coronary branches in MPR format as inputs. Disadvantages of both lesion- and branch-wise approaches are that errors on lesion-/branch-level are directly propagated to patient-level and that only local information is included in the network prediction. A case-wise CAD severity score is the Agatston score (Agatston et al., 1990), which in principle assesses the overall calcified plaque burden of a patient from non-contrast CT scans. This score can also be determined using machine learning methods (Wolterink et al., 2014; Lessmann et al., 2017; Cano-Espinosa et al., 2018). Our group recently proposed a case-wise approach to determine the CAD-RADS score (Denzinger et al., 2020b). It uses a hierarchical data representation of the whole coronary tree based on its anatomical sub-segments. For each of these sub-segments, features are extracted from the MPR volume stack with a CNN and combined with a global max pooling layer to predict the case-wise score. Based on the architecture and concepts presented in our previous work (Denzinger et al., 2020b), we present a more robust, streamlined and reproducible pipeline. Specifically, to ease reproducibility and simplify the pre-processing pipeline of our work we propose an architecture- and task-specific heuristic centerline labeling. Moreover,

we are leveraging the use of a severity-based label encoding, test time augmentation (TTA), model ensembling and reduced input dimensionality.

## 2. Data

Data is provided from a single site with CCTA scans acquired with the same scanner type. The number of patients (and samples) included is 2,902 with a fixed split of 1,926 used for training and 976 for testing. Within the test set, 131 patients have no CAD, 499 patients have non-obstructive CAD and 346 patients have obstructive CAD. The pre-processing is conducted as follows: after extracting the coronary centerlines using the method of (Zheng et al., 2013), MPR image stacks are extracted by interpolating planes orthogonal to the centerlines with a spacing of $(0.33 \times 0.33)\,\mathrm{mm}^2$ and a field of view (FOV) of $12 \times 12\,\mathrm{mm}^2$ for each centerline point with centerline points placed $0.25\,\mathrm{mm}$ apart. For these MPR image stacks, the Hounsfield unit (HU) value range is clipped to lie between $-300$ HU and $1,024$ HU with the resulting values being rescaled to a value range between 0 and 1.

## 3. Methods

### 3.1. Architecture

An overview of the used deep learning architecture is presented in Fig. 1, including an explanation of the individual steps.

### 3.2. Proposed Extensions

As the input for this network is either one or two orthogonal longitudinal views cut from the MPR volume stack at a specific angle $\alpha$ for each subsegment (cf. Fig. 1), the information used to predict the CAD-RADS score may vary. Therefore, the prediction may not be consistent with different angles, which it should be, given that for all angles the same biological information should be assessed. Our group showed in previous work (Denzinger et al., 2020a) that this problem can be partly solved by adding a second orthogonal view which still leaves some leeway for suboptimal angles especially when only one angle is considered during inference. To overcome this we leverage TTA averaging predictions for 16 views extracted for equally distributed angles between $[0, \pi]$ with the same angle for all segments. As the whole vessel information should be covered with this strategy, we additionally evaluate whether a single longitudinal view instead of two orthogonal longitudinal views suffices. Also, we propose to use model ensembling to lower uncertainty introduced by the network training converging to different local optima. In our prior work (Denzinger et al., 2020b), the prediction of the CAD-RADS score is transformed from a classification to a regression task and the network trained with a mean squared error (MSE) loss. This leads to all classes being weighted equally and the loss not depending on the individual class and how well this class has been learned already. To address this we suggest to use the following label encoding (Niu et al., 2016): $y_i^k = 1$ if $i \leq k$, $y_i^k = 0$ otherwise. Therefore, label vectors $\mathbf{y}^k$ belonging to class $k$ are created, with $i$ denoting the index of the entry in the label vector (e.g. CAD-RADS 2 is encoded as (1,1,1,0,0,0)). With this, we transform the regression task to a multi-label problem, which enables the use of a cross-entropy loss with

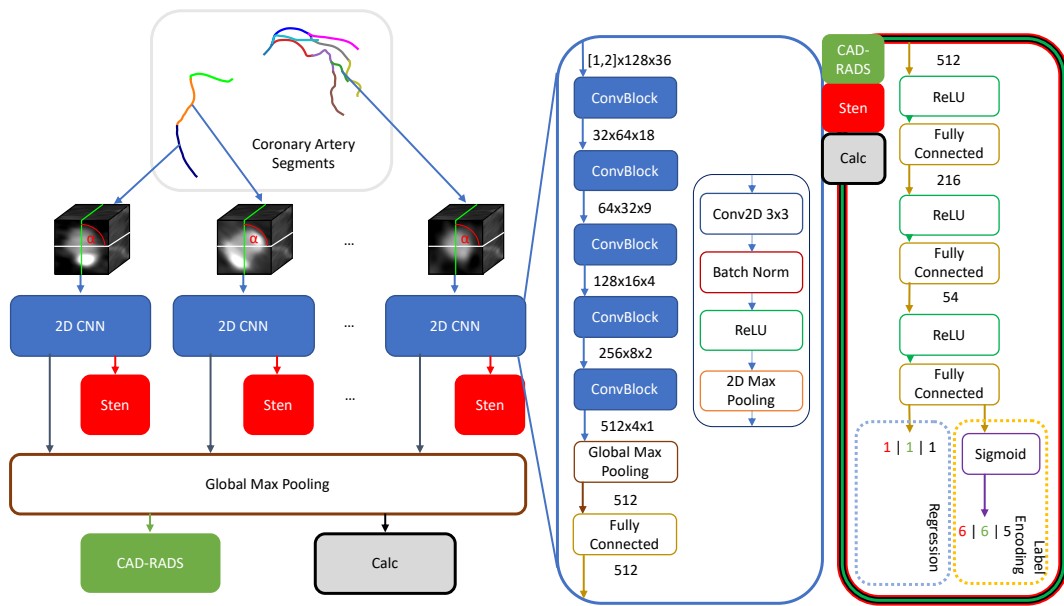

Figure 1: Overview of the architecture. For each labeled subsegment an MPR volume stack is computed and for one arbitrary angle $\alpha$ around the centerline, a longitudinal slice or two orthogonal longitudinal slices are extracted. The slices of all segments are fed into the same 2D CNN. The resulting feature representation is further processed by a multi-layer perceptron (MLP) for each segment to classify the stenosis grade and globally max pooled. The global feature representation is fed into two MLPs predicting the overall calcification (denoted as Calc and determined as a binned version of the Agatston score according to (Rumberger and Kaufman, 2003)) and the CAD-RADS grade. The output of the network is either one scalar value in case of regression or 5-6 sigmoidal outputs in case the labels are encoded as described in Section. 3.2.

sigmoidal predictions. During inference the raw predictions are summed over all outputs to get a cumulative probability and binned according to (Denzinger et al., 2020b).

### 3.3. Centerline Labeling

Furthermore, in the pipeline described in Reference (Denzinger et al., 2020b), the coronary tree was subdivided using the method proposed by (Gülsün et al., 2014) and the resulting segments were interpolated to one common length. With this a reasonable input to the network is obtained which may, however, yield obscured segments. Moreover, the extracted coronary tree usually exhibits more centerlines than defined in literature, since also small side branches are found by the centerline extraction algorithm of (Zheng et al., 2013). Furthermore, distal parts are usually less important and if a stenosis is present there it has less influence on thrombus formation or myocardial ischemia. Therefore, these should not necessarily have an impact on network prediction. Furthermore, even if the segment labels determined with the method of (Gülsün et al., 2014) are anatomically correct – which is

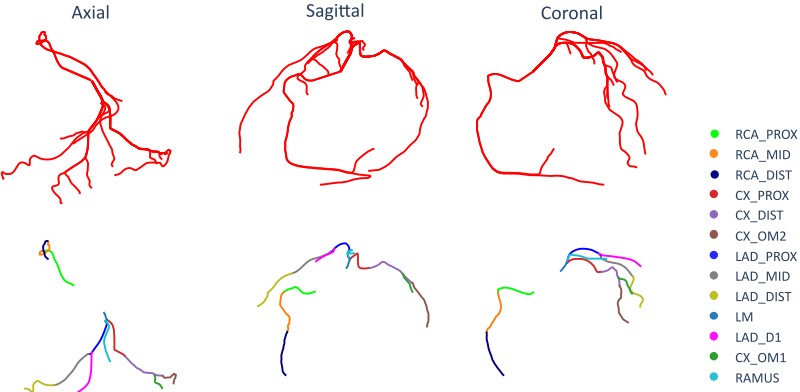

Figure 2: Centerlines before (top) and after (bottom) labeling. Note that centerline points inside the aorta originate from our data format and do not need to be labeled. Detected centerlines include the proximal, mid and distal part of the right coronary artery RCA ($\text{RCA}_{prox}$, $\text{RCA}_{mid}$ and $\text{RCA}_{dist}$), the left main segment ($LM$), the proximal, mid and distal part of the left artery descending (LAD) and left circumflex artery (CX) named $\text{LAD}_{prox}$, $\text{LAD}_{mid}$, $\text{LAD}_{dist}$ and $\text{CX}_{prox}$, $\text{CX}_{dist}$, $\text{CX}_{OM2}$, respectively, and the obtuse marginal (OM) artery of the CX $\text{CX}_{OM1}$ and the diagonal segment of the LAD, $\text{LAD}_{D1}$.

not always guaranteed – the segment image information is not directly transferable between patients due to the different segment lengths and potentially different supplied heart regions. We therefore propose an heuristic centerline labeling approach to solve previously mentioned problems with following notation [1]: let $\mathbf{C}$ be a set of centerlines $C$ consisting of centerline points $\mathbf{c} \in \mathbb{R}^3$. $\mathbf{c}_0$ is the first point of each centerline, which in our centerline format is always the center of the aorta with the first centerline points connecting the center of the aorta with the respective ostia. All centerlines end at their respective most distal point $\mathbf{c}_{n_c}$. This format leads to high redundancy in the centerlines with proximal parts often overlapping. An example of this and the abbreviations for the different segments are included in Fig. 2. Our heuristic pipeline is defined as follows: the set of centerlines can be subdivided into left $\mathbf{C}_l$ and right $\mathbf{C}_r$ centerlines by looking at their world coordinate direction starting from the center of the aorta. If hypothetically a different centerline-extraction algorithm outputs centerlines starting from the ostia, this initial step could be skipped. For the right centerline tree the longest segment $C_r^*$ is selected and, starting from the ostium, three subsequent segments of length 32 mm each are labelled as $\text{RCA}_{prox}$, $\text{RCA}_{mid}$ and $\text{RCA}_{dist}$, while the remaining vessel is excluded. For the left coronary tree, the bifurcation point $\mathbf{c}_b$ between the LAD and CX needs to be determined first. We detect $\mathbf{c}_b$ as the point where the centerlines of the left tree split most frequently. The LM is consequently labeled as the segment between the left ostium and $\mathbf{c}_b$. From $\mathbf{c}_b$ we calculate the directions of all centerlines containing this point as $\mathbf{c}_{b+10} - \mathbf{c}_b$. If there are **two** unique directions, the rightmost centerlines with this direction are defined as the LAD branch and the leftmost as

---

the CX branch. If there are **three** unique directions the branch between the others is labeled as RAMUS intermedius, which does not exist for all patients. The longest centerlines $C_{\mathrm{LAD}}^*$ and $C_{\mathrm{CX}}^*$ of the LAD and CX are divided into $\mathrm{LAD}_{prox}$, $\mathrm{LAD}_{mid}$, $\mathrm{LAD}_{dist}$ and $\mathrm{CX}_{prox}$, $\mathrm{CX}_{dist}$, $\mathrm{CX}_{OM2}$ respectively to obtain segments of lengths $32\,\mathrm{mm}$. Furthermore, for LAD and CX the centerlines $C_{\mathrm{LAD}}^{'}$ and $C_{\mathrm{CX}}^{'}$ which have the longest non-overlapping part to $C_{\mathrm{LAD}}^*$ and $C_{\mathrm{CX}}^*$ are selected. The $32\,\mathrm{mm}$ segments starting from the bifurcation between $C_{\mathrm{LAD}}^{'}$ / $C_{\mathrm{CX}}^{'}$ and $C_{\mathrm{LAD}}^*$ / $C_{\mathrm{CX}}^*$ respectively are labeled as $\mathrm{LAD}_{D1}$ and $\mathrm{CX}_{OM1}$. As the described heuristic approach does not aim to be absolutely anatomically correct and relies only on a small set of rules, it is consistent by design. Furthermore, it extracts segments of the same length, which eases the network training when it compares the segments of different patients. On the other hand, bifurcations do not only occur on the start and end of the segments, but also in the middle of the segment which leads to more diverse training data. Also, and maybe most importantly, it is simple and fast (around 350ms with a Intel(R) Xeon(R) CPU E5-2640 CPU).

### 3.4. Evaluation

For the evaluation, we keep our test set fixed while splitting our training data into five parts of approximately equal size in a stratified manner. We then use four of these parts as training and one as a validation set for five folds with the best model for each training with respect to the validation CAD-RADS score loss saved for evaluation. This setting is repeated five times for different seeds and splits for a total of 25 trained models for all configurations. Further hyperparameters were a stochastic gradient descent optimizer with a learning rate of 0.007/0.0007 for the label encoding/regression task respectively and a momentum of 0.99. We evaluate our different additions in form of an incremental study. First, the centerline labeling of the original approach is replaced with the one described in the section above. As the prediction of the network depends on the angle selected we will include the average results over all angles, the initial angle, and the angle with the retrospectively highest performance. Next, we use TTA taking the mean prediction over 16 angles equally distributed between $[0, \pi]$ with the same angle applied to all segments. This is followed by ensembling models and taking either the average prediction over the five folds of one seed or all 25 models. Finally, the label encoding is added, before testing whether a single view suffices.

## 4. Results and Discussion

As we have an ordinal classification task and are able to adapt the threshold depending on the desired ratio of sensitivity and specificity, we consider the area under the receiver operating curve (AUC) to be the most important metric. In general, we can see an incremental increase in performance with each improvement for the clinical question of rule-out (Table 1), hold-out (Table 2) and for predicting all six CAD-RADS grades (Table 3).
As we previously only reported the metrics for the views at a single angle in Reference (Denzinger et al., 2020b), it is hard to select which angle to choose for comparison. This task is also impacted by the fact that results differ at different selected angles. When averaging over all evaluated angles we get a mean AUC of 0.913 compared to 0.914 for the rule-out and 0.933 compared to a baseline of 0.923 for the hold-out case. However, looking at the

| Config/Metric | AUC | ACC | Sens | Spec | MCC |
|---|---|---|---|---|---|
| Baseline | 0.914 | 0.888 | 0.532 | 0.945 | 0.504 |
| + ~~TTA~~ $E_1$ ~~LE~~ 0 | 0.917±0.008 | 0.884±0.006 | 0.569±0.023 | 0.933±0.003 | 0.503±0.027 |
| + ~~TTA~~ $E_1$ ~~LE~~ * | 0.917±0.008 | 0.886±0.008 | 0.574±0.031 | 0.934±0.004 | 0.508±0.036 |
| + ~~TTA~~ $E_1$ ~~LE~~ ∀ | 0.913±0.006 | 0.880±0.004 | 0.555±0.016 | 0.931±0.002 | 0.486±0.019 |
| + TTA $E_1$ ~~LE~~ | 0.924±0.005 | 0.887±0.007 | 0.578±0.026 | 0.935±0.004 | 0.512±0.030 |
| + TTA $E_5$ ~~LE~~ | 0.932±0.001 | 0.890±0.002 | 0.591±0.007 | 0.937±0.001 | 0.527±0.008 |
| + TTA $E_{25}$ ~~LE~~ | 0.934 | 0.891 | 0.595 | 0.937 | 0.533 |
| + TTA $E_{25}$ LE | 0.941 | 0.895 | 0.611 | 0.940 | 0.550 |
| − TTA $E_{25}$ LE | **0.942** | **0.912** | **0.672** | **0.949** | **0.621** |

Table 1: Performance for the **rule-out task** for the different model configurations. Metrics are: the area under the receiver operating curve (AUC), accuracy (ACC), sensitivity (Sens), specificity (Spec), and Matthews correlation coefficient (MCC). "+/−" denotes whether two orthogonal or one single longitudinal view is fed into the CNN, "~~TTA~~/TTA" whether TTA is used, "$E_i$" the number of models ensembled, "~~LE~~/LE" whether labels are encoded as described in Section. 3.2 and "0/*/∀" whether the views extracted for the first, retrospectively best or all evaluated angles were considered. Baseline refers to the results reported in Reference (Denzinger et al., 2020b).

| Config/Metric | AUC | ACC | Sens | Spec | MCC |
|---|---|---|---|---|---|
| Baseline | 0.923 | 0.860 | 0.891 | 0.802 | 0.692 |
| + ~~TTA~~ $E_1$ ~~LE~~ 0 | 0.932±0.003 | 0.854±0.006 | 0.887±0.005 | 0.794±0.009 | 0.680±0.014 |
| + ~~TTA~~ $E_1$ ~~LE~~ * | 0.937±0.004 | 0.860±0.007 | 0.892±0.005 | 0.803±0.009 | 0.695±0.015 |
| + ~~TTA~~ $E_1$ ~~LE~~ ∀ | 0.933±0.004 | 0.856±0.004 | 0.888±0.003 | 0.797±0.006 | 0.686±0.010 |
| + TTA $E_1$ ~~LE~~ | 0.940±0.004 | 0.861±0.005 | 0.893±0.003 | 0.804±0.007 | 0.697±0.011 |
| + TTA $E_5$ ~~LE~~ | 0.943±0.000 | 0.860±0.002 | 0.892±0.001 | 0.803±0.003 | 0.695±0.005 |
| + TTA $E_{25}$ ~~LE~~ | 0.943 | 0.861 | 0.892 | 0.803 | 0.696 |
| + TTA $E_{25}$ LE | 0.944 | 0.861 | 0.892 | 0.803 | 0.696 |
| − TTA $E_{25}$ LE | **0.950** | **0.877** | **0.905** | **0.827** | **0.731** |

Table 2: Performance for the **hold-out question** for the different model configurations. Abbreviations as in Table 1.

angle with the best overall performance or the initial angle as an example, the performance is better than the baseline performance. This also nicely demonstrates why TTA is crucial. With TTA, a clear improvement in general performance is observed. This is easily explained by the fact that lesions cannot be missed by an unfortunate angle anymore. Ensembling multiple models leads to another performance boost, with an obvious improvement in stability when observing the decrease in standard deviation as the metric. Our proposed label encoding results in no improvement for the hold-out case, as the class balance is less severe in this case. However, for the rule-out case, an improvement from an AUC of 0.934 to

| Config/Metric | ACC | Sens | Spec | MCC |
|---|---|---|---|---|
| Baseline | 0.840 | 0.904 | 0.520 | 0.424 |
| + ~~TTA~~ $E_1$ ~~LE~~ $0/^*$ | $0.839{\pm}0.005$ | $0.904{\pm}0.003$ | $0.518{\pm}0.017$ | $0.422{\pm}0.021$ |
| + ~~TTA~~ $E_1$ ~~LE~~ $\forall$ | $0.834{\pm}0.004$ | $0.900{\pm}0.002$ | $0.504{\pm}0.014$ | $0.405{\pm}0.017$ |
| + TTA $E_1$ ~~LE~~ | $0.841{\pm}0.005$ | $0.904{\pm}0.003$ | $0.522{\pm}0.017$ | $0.426{\pm}0.020$ |
| + TTA $E_5$ ~~LE~~ | $0.842{\pm}0.001$ | $0.905{\pm}0.000$ | $0.525{\pm}0.004$ | $0.430{\pm}0.005$ |
| + TTA $E_{25}$ ~~LE~~ | 0.844 | 0.906 | 0.532 | 0.438 |
| + TTA $E_{25}$ LE | 0.845 | 0.907 | 0.535 | 0.442 |
| − TTA $E_{25}$ LE | **0.859** | **0.916** | **0.578** | **0.493** |

Table 3: Performance for the **six-class problem** averaged over all classes for the different model configurations. Abbreviations as in Table 1.

0.941 is observed. This illustrates that this change improves differentiation of less frequent classes. Finally, we tested decreasing the dimensionality by only having a single longitudinal view combined with TTA for each segment as an input for the network. This yielded far better results for all targets. A possible explanation for this may be the increased training stability that we observed and that the same information is fed to the system due to TTA. Moreover, beforehand the ordering of the two orthogonal longitudinal slices led to different results as different features were extracted for each, which should not be of relevance for the targets at hand. Especially the metrics for the six-class problem benefited the most from this change.

## 5. Conclusion

In this paper, we improve the automatic deep learning-based assessment of patients regarding the CAD-RADS score. We propose the use of TTA, model ensembling, task-specific label encoding, and reduced model input dimensionality for this task. Moreover, we introduce a novel task-specific heuristic centerline labeling approach, which by itself does neither lead to improved nor worse performance. However, it is easy to implement and makes the whole model pipeline easier to reproduce, while being theoretically more robust to technical variations due to its heuristic nature. Overall, we improve previously reported performance on the data set at hand: the accuracy for the six-class problem is increased to 0.859 from 0.840 and the AUC for the rule-out case to 0.942 from 0.914. For the hold-out case, we were able to reach an AUC of 0.950 compared to a previously reported 0.923. Further steps for this method are to apply it to data at different sites and/or scanner types.

**Disclaimer:** The methods and information here are based on research and are not commercially available.

**Acknowledgement:** K.B. gratefully acknowledges the support of the project "Dhip campus - bavarian aim".

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
