# OpenReview forum: "CAD-RADS Scoring using Deep Learning and Task-Specific Centerline Labeling"
_MIDL.io/2022/Conference — MIDL 2022_

### Official Review · Reviewer_38X6 · 2022-01-23

**Confidence:** 3
**Preliminary Rating:** 4
**Recommendation:** Poster

**Summary:**

The paper presents an extension of a method for automatic CAD-RADS scoring in CT angiography that has previously been published by the same authors. The method feeds multi-planar reformations of coronary artery segments (stretched-out vessels) into a 2D CNN and combines the output for the individual segments using global max pooling to predict a patient-level CAD-RAD score. This paper describes a different strategy for dividing the arteries into segments and adds test-time augmentation, model ensembling and an alternative label representation. Experiments with a single-center dataset with almost 3k scans show that these modifications result in better performance compared with the original method.

**Strengths:**

A strength of this paper is the large dataset of almost 3000 CCTA scans. The introduced modifications with respect to the author’s previous publication are generally well motivated, and it is certainly interesting to see the effect that such relatively small modifications have on the overall performance of an already well-performing method. The paper is well written and for most parts easily to follow.

**Weaknesses:**

The dataset is limited to a single center and a single scanner type, lacking an external test set. This is especially concerning because the paper investigates tweaks of an existing system to further improve its performance, but uses the test set for this. Whether these performance improvements are real and hold also for unseen data is not clear. Also, the paper leaves out details, for instance Figure 1 shows “Sten” and “Calc” outputs, but it is not described how these are used.

**Deanonymize Review:**

no

**Detailed Comments:**

The title does in my opinion not really reflect the content of the paper. Moreover, the abstract mentions “a patient” and in the following refers to this patient as “he” – even though the majority of patients with cardiovascular problems are male, please consider using gender neutral language instead.

**Final Rating After The Rebuttal:**

4: Weak Accept

**Justification Of The Final Rating:**

Taking the author's replies into account, my initial rating of this paper should have been "Borderline" with an upgrade now to "Weak Accept" - the paper clearly has some limitations (mostly an extension of a previously published paper, single center dataset) but would in my opinion be a good contribution to MIDL.

**Paper Type:**

methodological development

**Questions To Address In The Rebuttal:**

For test-time augmentation, how are the 16 angles chosen? Are these the same for all scans, for all segments in a scan, or do they differ per segment? Are they evenly distributed or are they completely random? How does this differ from “all evaluated angles”?

How was the reference standard obtained and which kind of labels are used for each scan / segment? How are the “Calc” and “Sten” outputs of the network used?

In Figure 1, the right-most box, is either the regression or the sigmoid output used, or is the sigmoid output used in addition to the regression output?


**Special Issue:**

no

---

### Official Review · Reviewer_tSNx · 2022-01-23

**Confidence:** 5
**Preliminary Rating:** 2

**Summary:**

The authors present a method to classify CT angiography scans as having no coronary artery disease (CAD), mild disease or severe disease. Briefly, the authors generate planar reformatted images from the coronary arteries centerlines and classify the case from them using convolutional neural networks. To improve performance the authors use several strategies: a) to turn the 3D data into 2D for the CNN b) use of test-time augmentation c) use of model ensembles and d) use a heuristic method to label the coronary arteries centerlines into canonical segments. Performance improves from their prior work in the exclusion of CAD or in the detection of severe CAD.

**Strengths:**

The authors address an important problem and use many optimization methods to improve the performance. The experiments are thoroughly done and explore the relevance of each of the methods used to improve the performance.

**Weaknesses:**

The literature review is incomplete. There is a lot of work related to the automated computation of the Agatston score, another measurement to evaluate the severity of CAD. Such work should be included in the introduction and discussion. Also methods that do not extract centerlines should be addressed. Examples:
https://www.ncbi.nlm.nih.gov/pmc/articles/PMC6095680/
https://ieeexplore.ieee.org/abstract/document/8094970?casa_token=IJQZds70YwgAAAAA:jdPWRr2US18k5XGgMpghVMtwBveGhtIl0smCew4FG1wQ5EePgtjR3Nu9Vtj84nza8ANkjK3iFQ
https://www.spiedigitallibrary.org/conference-proceedings-of-spie/9035/90350E/An-automatic-machine-learning-system-for-coronary-calcium-scoring-in/10.1117/12.2042226.short
It is unclear how good (or bad) is the method of Zheng et al. for centerline extraction. Being such method a cornerstone of the following processing, it is important to understand its performance. Similarly, while the heuristic method to assign labels to the centerlines seems well motivated, its performance is unknown. The paper would benefit of its evaluation in a subset of cases.
The problem is inherently 3-dimensional. The authors chose to use 2D convolutional networks by finding single or multiple planar reformats of the coronary arteries. Given the moderate size of the reformatted images (128x36 pixels, Fig. 1), one would assume the problem could be directly tackled in 3D.
It is unclear which angle is used to generate the image used to train the neural network.
The performance of the ensemble learning is doubtful since the values obtained when using 5 or 25 models falls within the standard deviation of using one model in several of the metrics.
The encoding of Eq. 1 needs to be better justified. Why not using a 1-hot encoding? In test time, what happens if a case has the labels roughly of (1,0,1,0,0,0). Would that be interpreted as CAD-RADS 2 or 0?
The convolutional networks used seem to be fairly simple. One wonders if more advanced architectures would yield better results.


**Deanonymize Review:**

no

**Detailed Comments:**

Hounsvield – Hounsfield (Section 2)

**Paper Type:**

validation/application paper

**Questions To Address In The Rebuttal:**

I would like the authors to address the weaknesses described above. Many of them are related to presentation and clarifications and do not require of further experimentation or methodological changes.

**Special Issue:**

no

---

### Official Review · Reviewer_qKHB · 2022-01-27

**Confidence:** 4
**Preliminary Rating:** 2
**Recommendation:** Poster

**Summary:**

The authors present an extension of their MICCAI 2020 paper on deep learning-based CAD-RADS scoring in coronary CTA scans. The extension of their previous work consists of adapting a new loss function, performing model ensembling, and integrating a test time augmentation approach. Ensembling and test time augmentation are shown to improve performance; the loss function further improves performance for the so-called rule-out scenario (CAD-RADS 0 vs. 1-5). The authors also propose a new labeling approach of coronary artery segments that does not affect prediction performance. Since the focus of MIDL is by definition on deep learning, I will focus on the respective aspects in the following.

**Strengths:**

**General aspects:**
The manuscript is in general well-structured and the language is appropriate. Prior work is adequately addressed.

**Strengths / main contributions:**
It is nice to see that fine-tuning the knobs of the authors' original approach and adding standard approaches like model ensembling further improves the performance of the original model. In general, the selected extensions/modifications are reasonable (as also reflected by the gain in performance) and the data set underlying the presented experiments is (for our community) relatively large, rendering the presented results reliable.


**Weaknesses:**

Given the focus of MIDL, I am somewhat irritated by the division of the paper: >1.5 pages for the adapted centerline labeling and < 1 page for the extensions of the deep learning system (+ 1 overall figure).
Since I am not 100% familiar with the data at hand, this led me to be unsure whether I understood the TTA aspect in full detail. In particular, on page 3, section 3.2, it is written that “longitudinal views cut from the MPR volume stack at a specific angle for each subsegment”. Which angle is specifically meant? Since this is the basis for TTA and thus a central part of the paper, I found it somewhat disappointing to have to read the manuscript with only a (perhaps wrong) gut feeling which angle might be meant. And based on my gut feeling, I was personally very surprised that the “one view” results are better than those of the “two views” (maybe indicating that the gut feeling is actually wrong).

Continuing the above comment, concerning the adapted loss function, how does the approach address/solve the MSE drawbacks mentioned on page 4? Where does the “severity-based” aspect come in that is mentioned in the abstract and the introduction (and never again thereafter)?

As far as I can see, it is not intended to make the models/source code and data publicly available.


**Deanonymize Review:**

no

**Detailed Comments:**

-	Introduction, first two sentences: Where is the (causal) relationship between CAD being a leading cause of death and developing algorithms to be key to improving diagnosis and patient outcomes?
-	Introduction: manual grading as a time-consuming task and reader-dependent: Are there corresponding literature sources for this (and numbers)?
-	Figure 1: Why a calcification prediction? This is not mentioned at all in the manuscript text.
-	Tables: To which repeated runs correspond the standard deviations? (Did I miss it?)
-	Conclusion: “large margin” for AUC improvement from 0.84 to 0.859? I would suggest being a bit moderate.


**Final Rating After The Rebuttal:**

3: Borderline

**Justification Of The Final Rating:**

I thank the authors for their clarification and revision, which resulted in me changing my overall rating. However, due to the focus of the manuscript (as discussed in the review, the authors' rebuttal, and my response), I am still somewhat hesitant and will leave the decision to the AC.

**Paper Type:**

methodological development

**Questions To Address In The Rebuttal:**

The intention of the present paper is, in my opinion, to show that finetuning the authors’ MICCAI 2020 deep learning approach leads to improved performance. While the data sets used for evaluation are (given the numbers in the two manuscripts; MICCAI numbers as given in the corresponding arXiv preprint) not 100% identical for the present contribution and MICCAI, the numbers are similar and a comparison seems to be appropriate. I therefore understand the intention to publish the recent results. Given the high standards of MIDL, it is, however, not clear whether the presented extensions are sufficient for acceptance.
In the rebuttal, I would particularly like to ask the authors to explain what is meant by “angle” (see comments above) and, in case of acceptance, to add a respective explanation to the manuscript.

**Special Issue:**

no

---

### Official Review · Reviewer_2QC5 · 2022-01-27

**Confidence:** 5
**Preliminary Rating:** 5
**Recommendation:** Oral, Poster

**Summary:**

This paper extends and improves a previously published method by the same author group to find and analyze the coronaries in CTA images and assign a CAD-RADS score.

---------------------------------------------------------------------------------------------------------------------------------------------

**Strengths:**

The paper is clearly written. The work is, as far as I can judge without having access to the code and the data, of good quality. The experiments described are sensible. The data set used is relatively large and seems to be of good quality.

**Weaknesses:**

The paper is an extension of previous work, somewhat limiting novelty. The code is not publicly available. The data is single center and single scanner, and thus the method is trained and tested on data from the same scanner. This makes it impossible to say anything about generalizability. The data is not publicly available. Thus it is impossible to claim that this "we improve previously reported state-of-the- art by a large margin". In fact, previously published methods may outperform this method by a large margin. With the data and the method kept proprietary, we will never know.

**Deanonymize Review:**

yes

**Detailed Comments:**

The conclusion that the 2 orthogonal views offer no advantage after adding the rotations is not surprising. In the original method without TTA using multiple views makes sense otherwise you simply ignore a lot of the available data. With TTA you analyze all data, if you use enough rotations. The original approach of feeding two orthogonal views to a network that starts with a convolution layer is inherently weird. A convolution makes sense because it is a local operation and neighboring pixels describe local structure, but the corresponding pixels in orthogonal views are not close in proximity so convolution is not a logical operation (it may still work to a certain extent of course).

**Final Rating After The Rebuttal:**

5: Strong Accept

**Justification Of The Final Rating:**

I appreciate the author’s answers and their decision to share some of the code.
—————————————————————————————————————————————————————————————————————————————————————————————————————————————————————————-

**Paper Type:**

validation/application paper

**Questions To Address In The Rebuttal:**

It would be outstanding if the code and data were shared with a permissive license.
--------------------------------------------------------------------------------------------------------------------------

**Special Issue:**

no

---

### Meta-Review · Area_Chair_Ti47 · 2022-02-18

**Recommendation:** Accept (Poster)
**Confidence:** 4

**Metareview:**

The reviewers' opinions varied a lot for this paper, so I investigated a bit further to understand all perspectives and give a fair recommendation.

As some of the reviewers also pointed out, the paper is a well-conducted and well-written extension of prior work, evaluated on a large but private dataset. The reviewers agreed that while sharing the data would be preferrable, this could not be expected from the authors.

One reviewer initially gave weak score and omitted revising it after the authors responded (in my opinion appropriately) to all their comments. I'm inclined to largely dismiss this preliminary rating. For one of the reviewers, the main reason for rejecting the paper would be that the paper focuses more heavily on a processing pipeline rather than novel deep learning techniques. The reviewer therefore feared the submission might not be as well-aligned with the scope of the MIDL conference as other submissions. However, I believe preparatory processing steps are integral parts of deep learning systems and should be within the scope of MIDL.

Given all these considerations, I believe the paper merits publication and recommend to accept it.

---

### Decision · Program_Chairs · 2022-02-28

Accept